

# High cyclic GMP-AMP synthase and stimulator of interferon genes in cholangiocarcinoma suggest their potential as targets for treatment

Han Ni Zin Myint[1,*], Paksiree Saranaruk[1,2,*], Natnicha Paungpan[1,2,3], Sakda Waraasawapati[4], Chawalit Pairojkul[4], Saowaluk Saisomboon[1,2,3], Nutsara Settha[1], Jatuporn Pattanachot[1], Seiji Okada[3], Wunchana Seubwai[5,6], Amnat Kitkhuandee[6,7], Kanlayanee Sawanyawisuth[1,2,6] and Kulthida Vaeteewoottacharn[1,2,3,6]

[1] Department of Biochemistry, Faculty of Medicine, Khon Kaen University, Khon Kaen, Thailand
[2] Cholangiocarcinoma Research Institute, Khon Kaen University, Khon Kaen, Thailand
[3] Division of Hematopoiesis, Joint Research Center for Human Retrovirus Infection and Graduate School of Medical Sciences, Kumamoto University, Kumamoto, Japan
[4] Department of Pathology, Faculty of Medicine, Khon Kaen University, Khon Kaen, Thailand
[5] Department of Forensic Medicine, Faculty of Medicine, Khon Kaen University, Khon Kaen, Thailand
[6] Center for Translational Medicine, Faculty of Medicine, Khon Kaen University, Khon Kaen, Thailand
[7] Department of Surgery, Faculty of Medicine, Khon Kaen University, Khon Kaen, Thailand
* These authors contributed equally to this work.

Corresponding author
Kulthida Vaeteewoottacharn,
kulthidava@kku.ac.th

## ABSTRACT

**Background:** Cancer remains the leading cause of death worldwide. Chromosomal instability (CIN) is one hallmark of cancer. Micronuclei (MN) is an observable outcome of CIN. The role of cytosolic MNs or DNAs in activating an immune response *via* a cyclic guanosine monophosphate–adenosine monophosphate (GMP-AMP) synthase (cGAS)-a stimulator of interferon genes (STING) signaling cascade is established. However, the information regarding the cGAS-STING pathway in cholangiocarcinoma (CCA) is limited. This study aimed to determine cGAS-STING-related molecules in CCA and reveal their clinical importance.

**Methods:** MNs in CCA cell lines were demonstrated by nuclear staining and cGAS, STING, nuclear factor kappa B (NF-κB) p65 were determined by Western blot analysis. Expressions of cGAS-STING-related mRNAs in gastrointestinal cancers were analyzed using Gene Expression Profiling Interactive Analysis (GEPIA) based on The Cancer Genome Atlas (TCGA) database. The expressions of cGAS, STING, and NF-κB p65 in clinical CCA samples were determined by immunohistochemistry (IHC) staining. The survival analyses were conducted using Kaplan-Meier survival analysis with a log-rank test, and the correlations between the targeted protein expression levels and clinical parameters were examined using Pearson's Chi-square test. Furthermore, the Cox proportional hazard regression model was utilized for multivariate analysis. The expression levels of *CGAS, STING*, and *RELA* were analyzed using two public datasets of CCA. The relationship between key mRNAs and related immune cell infiltration was analyzed using the Tumor Immune Estimation Resource (TIMER).

**Results:** MNs generated during cytokinesis were demonstrated in the CCA cell lines and were positively correlated to STING level. GEPIA analysis revealed that members of the cGAS-STING pathway were significantly increased in GI cancers, particularly CCA and pancreatic cancer. IHC confirmed high cGAS and STING in CCA tissues, while NF-κB p65 showed variable expression. High NF-κB p65 was correlated with shorter patient survival, and larger tumor size. High NF-κB p65 contributed to an increased hazard ratio of 1.679 (95% confidence interval [1.074–2.626]). TIMER analysis demonstrated *CGAS* and *STING* were positively correlated with neutrophil, γδ T cell, and CD8$^+$ T cell infiltration, while *STING* and *RELA* were associated with higher B cell infiltration. High *CGAS*, *STING*, and *RELA* were related to increased myeloid dendritic cell infiltration.

**Conclusions:** The high number of MNs in CCA cell lines emphasize their importance. Elevated cGAS, STING, and NF-κB p65 in CCA tissues highlight the significance of this pathway. While cGAS and STING showed no direct prognostic value, NF-κB p65 was identified as a prognostic predictor. Altogether, the opportunity of cGAS and STING targeting for CCA treatment and the predictive character of NF-κB p65 are proposed, and further exploration is recommended.

# INTRODUCTION

Cancer is one of the leading causes of death worldwide, with an estimated 20 million new cases and 9.7 million cancer-related deaths reported in 2022 (*Bray et al., 2024*). Chromosomal instability (CIN), characterized by structural and numerical chromosomal abnormalities, is a hallmark of cancer (*Tanaka & Hirota, 2016*). Micronuclei (MNs) are one well-known CIN biomarker. They are released from the nucleus into the cytoplasm of mis-segregated chromosomes (*Bakhoum et al., 2018*). The functions of MNs in the inflammation promotion are established (*Dou et al., 2017*). MNs and cytosolic DNAs can activate the cyclic guanosine monophosphate–adenosine monophosphate (GMP-AMP) synthase (cGAS)-stimulator of interferon genes (STING) pathway, thereby triggering immune responses (*Dou et al., 2017*; *Mackenzie et al., 2017*). The outcomes of the cGAS-STING pathway in cancers are target-dependent (*Bakhoum & Cantley, 2018*; *Khoo & Chen, 2018*; *Ng et al., 2018*). cGAS-STING-stimulated type 1 interferons (IFNs) promote cancer removal, whereas cGAS-STING-induced nuclear factor kappa B (NF-κB) activation and pro-inflammatory cytokine production enhance carcinogenesis and cancer metastasis (*Bakhoum et al., 2018*; *Beernaert & Parkes, 2023*). Therefore, exploring the cGAS-STING pathway in clinical samples might be useful for understanding the nature of cancer.

Cholangiocarcinoma (CCA), or bile duct cancer, is a significant health problem in East Asia and the Greater Mekong subregion (*Banales et al., 2020*). The early detection of CCA is challenging due to its nonspecific symptoms and varied clinical manifestations (*Banales et al., 2020*). The gold standard treatment of CCA is curative surgery; however, it is limited to patients with early-stage disease (*Vogel et al., 2023*). The proportion of CCA patients

eligible for curative surgery varies by geography; for instance, 58.7% of US-based Surveillance, Epidemiology, and End Results (SEER) registry patients diagnosed in stage I and II, while 43.3% of Thai CCA patients were classified in the same stages (*Kim et al., 2017*; *Sa-Ngiamwibool et al., 2022*). Recommended therapies for advanced CCA largely depend on the actionable targets observed in cancers (*Banales et al., 2020*; *Vogel et al., 2023*). Clinical trials using anti-PD-L1 plus gemcitabine and cisplatin have shown promising results (*Burris et al., 2024*; *Finn et al., 2024*). Hence, the novel treatment regulating immune response might be useful.

The relationship between chronic inflammation and CCA development or progression is well-documented and independent of significant risk factors (*Labib, Goodchild & Pereira, 2019*). Proinflammatory cytokines, including IL-6, IL-8, and IL-10, play essential roles in CCA promotion and progression (*Li et al., 2022*; *Sripa et al., 2012*). NF-κB-mediated IL-6 and IL-8 expressions in CCA have been revealed (*Dana et al., 2017*). Moreover, CIN and aneuploidy are common features of CCA (*Dalmasso et al., 2015*; *Jusakul et al., 2017*). However, the information regarding CIN and cGAS-STING pathway in CCA is limited. Thus, the current study aimed to determine the expression of cGAS, STING, and its downstream molecule, NF-κB, in Thai CCA tissues and emphasize their clinical significance.

We revealed MNs in CCA cell lines. These MNs were detected during mitosis and related to STING levels. Using data from public databases, we highlighted the significance of the cGAS-STING pathway in CCA and pancreatic cancer. Immunohistochemistry (IHC) staining and statistical analysis revealed the importance of cGAS and STING as mutual targets for CCA treatment and NF-κB as a prognostic predictor. Altogether, this study provided a better understanding of cGAS-STING-NF-κB roles and their implications in CCA.

## MATERIALS AND METHODS

### Human CCA tissues
Ninety-five CCA tissues were collected from patients diagnosed with CCA and treated at Srinagarind hospital, Khon Kaen University. The paraffin-embedded tissues were obtained from the specimen bank of Cholangiocarcinoma Research Institute, Faculty of Medicine, Khon Kaen University. Written informed consent was obtained from all patients. The selection and exclusion criteria were as previously reported (*Saranaruk et al., 2023*). The study procedure was approved by the Ethics Committee for Human Research of Khon Kaen University based on the Declaration of Helsinki (HE641574 and HE651124).

### CCA cell lines
Three CCA cell lines, KKU-055, KKU-213A (*Sripa et al., 2020*), KKK-D068 (*Vaeteewoottacharn et al., 2019*) cells were obtained from the Japanese Collection of Research Bioresources Cell Bank (JCRB) (Osaka, Japan). Cells were cultured in Dulbecco's modified Eagle's medium (DMEM) supplemented with 10% fetal bovine serum (FBS) and 1% antibiotic-antimycotic in a humidified incubator at 37 °C with 5% $CO_2$. All cell

culture-related reagents were purchased from Gibco (Life Technologies, Grand Island, NY, USA).

## Antibodies and reagents

The antibodies were obtained from the following sources: rabbit anti-cGAS (#26416-1-AP), anti-STING (#19851-1-AP), and mouse anti-NF-κB p65 (#66535-1-Ig) were from Proteintech (IL, USA), mouse anti-NF-κB p65 (F6) from Santa Cruz Biotech (CA, USA), mouse anti-GAPDH from EMD Millipore (Darmstadt, Germany), anti-mouse IgG-conjugated with horseradish peroxidase (HRP) (62-6520) from Invitrogen (CA, USA), anti-lamin A/C (4777) and anti-rabbit IgG HRP (7074S) from Cell signaling Technology (MA, USA), anti-mouse (K4001) and anti-rabbit (K4003) EnVision HRP-conjugated antibodies were from DAKO (Glostrup, Denmark).

3,3′-Diaminobenzidine Tetrahydrochloride hydrate (DAB) (D5637) were bought from Sigma-Aldrich; Immunobilion$^{®}$ Forte Western HRP substrate (WBLUF0500) from EMD Millipore; Quick Start$^{TM}$ Bradford 1x Dye Reagent (5000205) from Bio-Rad; Hoechst33342 (H3570) was from Invitrogen.

## mRNA expression analysis

To investigate the expression of the cGAS-STING-related molecules, including *CGAS, STING, TANK-binding kinase 1 (TBK1), interferon regulatory factor 3 (IRF3), NF-κB p65 (RELA), NF-κB p50 (NFKB1), interleukin-6 (IL6), tumor necrosis factor (TNF), interferon alpha 1 (IFNA1)*, and *IFN beta 1 (IFNB1)*, in gastrointestinal cancers compared with comparable normal tissues, we assessed using the Gene Expression Profiling Interactive Analysis (GEPIA, http://gepia.cancer-pku.cn/index.html), an online server for cancer and normal gene expression profiling analysis based on The Cancer Genome Atlas (TCGA) database (*Tang et al., 2017*).

The expression levels of *CGAS, STING*, and *RELA* in CCA tissues were retrieved from Gene Expression Omnibus (GEO: https://www.ncbi.nlm.nih.gov/gds/), including GSE26566 (*Andersen et al., 2012*), and GSE76297 (*Chaisaingmongkol et al., 2017*). The GSE26566 dataset included 104 CCA and 71 normal samples. The cohort 91 CCA and 92 normal adjacent tissues were obtained from GSE76297. Log$_2$ mRNA expressions in cancer and normal tissues were compared.

## IHC staining

According to the standard guidance, the expression levels of targeted proteins in the formalin-fixed paraffin-embedded CCA tissues were determined using IHC staining using antibodies against cGAS (1:100), STING (1:200), and NF-κB p65 (1:100). The signal amplification was performed using the EnVision system-HRP system. The immunoreactivity was visualized by DAB.

The cGAS, STING and NF-κB p65 expression levels were assessed using H-scores (*Fitzgibbons et al., 2014*; *Saranaruk et al., 2023*). The patients were categorized into low- and high-expressed groups using median H-score value. Two independent evaluators and

a pathologist independently evaluated the H-scores while blinded to the clinical information.

## Hoechst 33342 staining

KKU-055, KKU-213A, and KKK-D068 cells were seeded at a density of $2 \times 10^4$ cells/well into a 24-well plate and incubated overnight. After washing with PBS, cells were fixed with 4% paraformaldehyde for 30 min, then stained with 1 µg/ml Hoechst33342 for 10 min. The micronuclei were observed using a confocal laser scanning fluorescence microscope (Carl Zeiss confocal microscope LSM 800, Jena, Germany) with objective 40X magnification. More than 500 cells from randomly 15–30 fields were counted and calculated for the percentage of MN-containing cells as follows: [numbers of cells with MNs/total cells]*100.

## Protein extraction and Western blot analysis

Total protein was extracted using RIPA lysis buffer (50 mM Tris-HCl pH 7.4, 150 mM NaCl, 1% NP-40, 1% Na-Deoxycholate, 0.1% SDS) with protease and phosphatase inhibitor cocktail. Nuclear proteins were prepared as described previously (*Seubwai et al., 2010*). Briefly, cells were washed and lyzed in a hypotonic solution. Nuclei were collected by centrifugation and lyzed by nuclear lysis buffer (50 mM HEPES-KOH, pH 7.9, 1.5 mM $MgCl_2$, 10 mM KCl, 0.1% NP-40, 0.5 mM DDT) containing protease and phosphatase inhibitors.

Protein concentrations were quantified by Bradford reagent. Proteins were subjected to SDS-PAGE under denaturation conditions and transferred to PVDF membrane. The detection of targeted proteins was performed using specific antibodies: anti-cGAS (1:2000) anti-STING (1:2000), anti-NF-κB p65 (1:1000), and anti-GAPDH (1:5000). Immunoreactivity was detected by Immunobilion®Forte western HRP substrate and visualized by an ImageQuant LAS 600 (GE Healthcare; Buckinghamshire, UK). Protein expression intensities were analyzed using ImageJ software version 1.53k (*Schneider, Rasband & Eliceiri, 2012*).

## Estimation of immune cell infiltration

The immune cell infiltration (ICI) in CCA tissue was analyzed by a web server, Tumor Immune Estimation Resource (TIMER2.0, http://timer.cistrome.org/), based on computational algorithms and TCGA database (*Li et al., 2020*). The correlations between *CGAS*, *STING*, and *RELA* mRNA expression levels and the infiltration levels of different immune cells were evaluated by Spearman's rank correlation coefficient (Rho) and plotted as scatter plots. The statistical significance of correlations is presented as a *p*-value.

## Statistical analysis

Statistical analyses were conducted using SPSS version 28.0.1.0 (SPSS Inc., IL, USA) and GraphPad Prism®8.0.2 Software (GraphPad Software Inc., La Jolla, CA, USA). Survival analyses between different groups were performed by Kaplan-Meier analysis with a log-rank test. The differences in the clinical features were compared using the Chi-square test. The multivariable Cox regression model was utilized to identify independent
clinicopathological characteristics linked to patient survival, followed by testing backward selection to avoid the influence of confounding factors (*Grant, Hickey & Head, 2019*). The Pearson correlation coefficient (r) was applied to measure a linear correlation between two variables.

The quantitative data were presented as mean ± SD from three independent experiments. Multiple comparisons of means were analyzed by ANOVA, followed by a student's t-test to identify differences between two groups. *p*-value less than 0.05 was considered statistically significant.

## RESULTS

### Micronuclei in CCA cell lines

CIN is one characteristic of CCA (*Dalmasso et al., 2015*; *Jusakul et al., 2017*); thus, the presence of MN was expected. MNs were detected in 3 CCA cell lines: KKU-055, KKU-213A, and KKK-D068. The results demonstrated that MNs were commonly visualized in CCA cell lines (Fig. 1A). Among them, KKK-D068 acquired the highest percentage of MN-containing cells, which was statistically significant when compared to the KKU-055 (Fig. 1B). Moreover, mis-segregated chromosome-derived MNs were generally observed in KKK-D068 cells (Figs. 1C, 1D).

The expression levels of cGAS, STING, and NF-κB p65 proteins were assessed by Western blot analysis. The results demonstrated that cGAS was highest in KKU-213A, and STING was elevated in KKK-D068 (Fig. 1E). NF-κB p65 was abundantly expressed in all three cell lines in both whole-cell lysates and nuclear fractions (Figs. 1E, 1F). Notably, only STING levels were correlated with MN formation in cells.

### cGAS-STING pathway-related molecules were elevated in CCA and pancreatic adenocarcinoma (PAAD)

GEPIA web-based tool was used to evaluate the mRNA expression of cGAS-STING pathway-related molecules in gastrointestinal cancers, including CCA, PAAD, liver hepatocellular carcinoma (LIHC), colon adenocarcinoma (COAD), esophageal carcinoma (ESCA), rectum adenocarcinoma (READ), and stomach adenocarcinoma (STAD). *CGAS, STING, TBK1, IRF3, NF-κB p65* (*RELA*), and *NF-κB p50* (*NFKB1*), *interleukin-6* (*IL6*), *tumor necrosis factor* (*TNF*), *interferon alpha 1* (*IFNA1*), *IFN beta 1* (*IFNB1*) were selected as the candidate molecules. A schematic representation of the cGAS–STING signaling pathway and its related molecules is shown in Fig. 2A. The results revealed that *CGAS* was highly detected in CCA, PAAD, ESCA, and STAD when compared to the normal counterparts, while *STING, TBK1, IRF3, RELA*, and *NFKB1* were significantly upregulated in CCA and PAAD (*p* < 0.05). A significant increase *in IL6* was observed in PAAD and ESCA (Fig. 2B). These results indicated that cGAS-STING pathway-related molecules were mostly induced in CCA and PAAD. The increased cGAS-STING pathway-related molecules observed in the public database prompt us to determine protein expression in the clinical CCA samples.

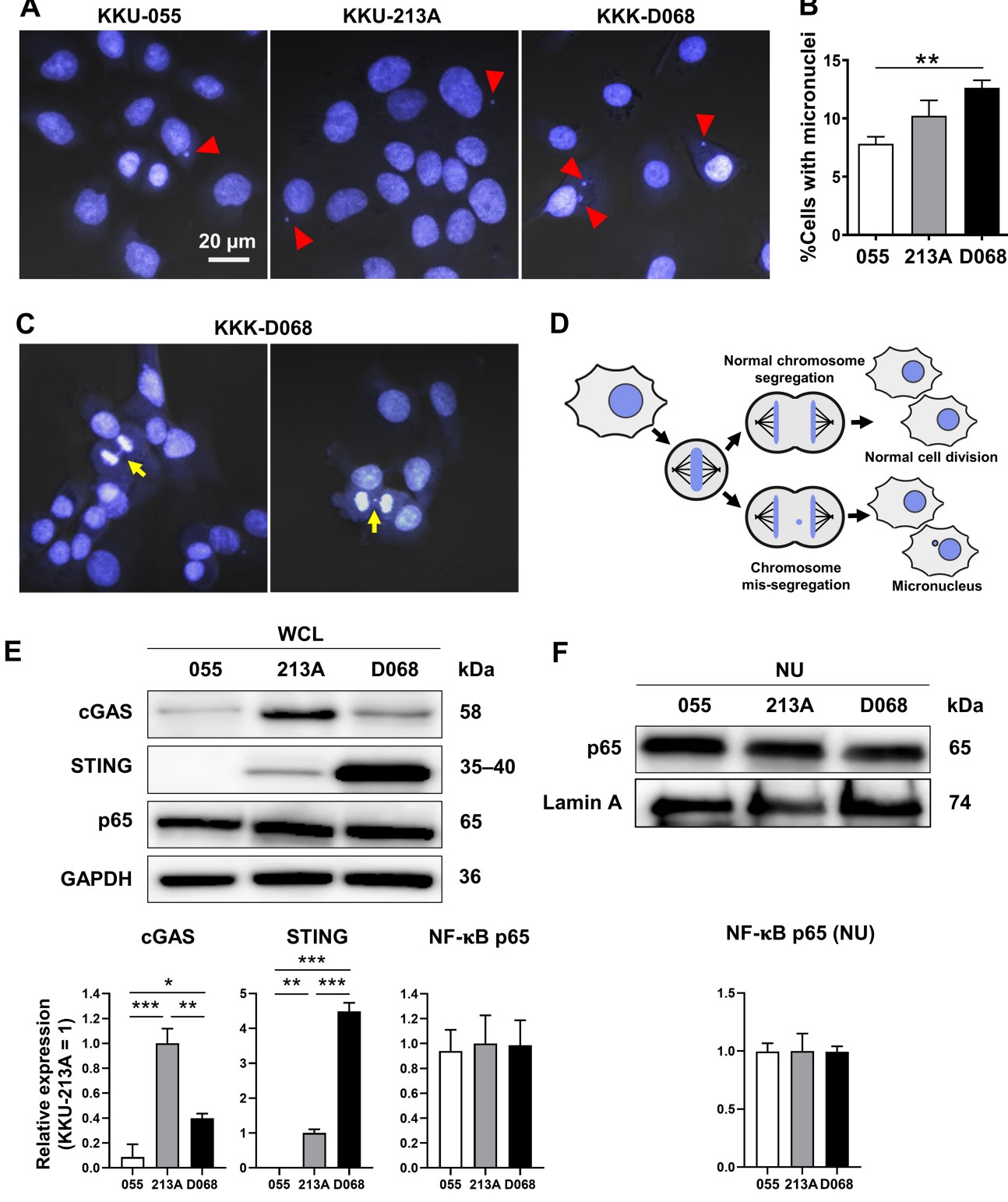

**Figure 1 Micronuclei, cGAS, STING, and NF-κB p65 protein expression levels in CCA cell lines.** (A) Micronuclei (MNs) in KKU-055, KKU-213A, and KKK-D068 were demonstrated by Hoechst33342 staining. Red arrowheads indicate MNs. (B) The graph represents % MNs observed in KKU-055 (055), KKU-213A (213A), and KKK-D068 (D068). (C) MNs observed in mitotic KKK-D068 cells (yellow arrow). (D) Diagram

**Figure 1 (continued)**
showing MNs-generated during chromosome mis-segregation. (E) Representative expression of cGAS, STING, and NF-κB in whole cell lysate (WCL) and (F) nuclear extract (NU) of KKU-055 (055), KKU-213A (213A), and KKK-D068 (D068). GADPH and lamin A serve as loading controls. Graphs present relative levels of cGAS, STING, and NF-κB in three cell lines from three independent experiments. Expression level in KKU-213A = 1. *$p < 0.05$, **$p < 0.01$, ***$p < 0.001$. 

## Demographic characteristics of CCA

A total of 95 CCA samples were selected to evaluate cGAS–STING pathway-related molecule expression. The patient ages ranged from 37 to 76 years, with a median age of 56. The cohort comprised 62 males and 33 females (a male-to-female ratio is 3.8:2). Anatomically, 33 cases (34.7%) were perihilar CCA and 62 cases (65.3%) were intrahepatic CCA. Fifty-two samples (54.7%) were non-papillary histological subtypes. CCA staging was performed according to the 7th edition of the American Joint Committee on Cancer (AJCC) staging system (*Edge & Compton, 2010*); 52 cases (54.7%) were stage IV. Tumor sizes were reported in 78 cases (82.1%); 41 cases (52.6%) had tumors ≥7 cm, while 37 cases (47.4%) had tumors <7 cm (Table 1). The overall survival time, calculated from the date of surgery to death, ranged from 30 to 2,593 days, with a median survival time of 377 days. Surgical interventions included both curative and palliative procedures.

## High expression of cGAS, STING, and differential expression of NF-κB p65 in CCA tissues

To demonstrate the expression of cGAS-STING pathway-related molecules in the clinical CCA samples, cGAS, STING, and NF-κB were selected as the representative proteins. Protein expression was assessed by IHC staining and semi-quantitatively assessed using H-scores. The representative IHC images are presented in Fig. 3A and H-score distributions are shown in Fig. 3B. The median H-scores of cGAS and STING were 300, indicating consistently high expression across the cohort. In contrast, the median H-score for NF-κB p65 was 135, reflecting variable expression.

Kaplan-Meier survival analyses using the log-rank test showed no significant correlation between the expression levels of cGAS, STING, and NF-κB p65 and overall patient survival (Figs. S1A–S1C). However, Pearson's chi-square test revealed that cGAS and STING expression levels were significantly higher in perihepatic CCA compared to intrahepatic CCA (Table 1, $p = 0.046$ and $p = 0.030$). Additionally, elevated NF-κB p65 was observed more frequently in tumors ≥7 cm ($p = 0.043$).

The non-papillary CCA is more aggressive and associated with poorer outcomes (*Saranaruk et al., 2023*; *Zen et al., 2006*). Therefore, survival analyses were conducted separately for papillary and non-papillary subtypes. In non-papillary CCA, tumor stage IV and high NF-κB p65 were correlated with shorter survival of the patients (Figs. 3C, 3D). Patients with stage IV CCA had a mean overall survival of 344 ± 96 days, compared to 627 ± 282 days for those with CCA stage I–III ($p = 0.042$). Similarly, patients with high NF-κB p65 expression had a significantly shorter mean survival time (257 ± 79 days) than those with low expression (638 ± 212 days; $p < 0.001$). In contrast, the expression levels of cGAS and STING in non-papillary CCA (Figs. S2A, S2B), as well as the expression of all three

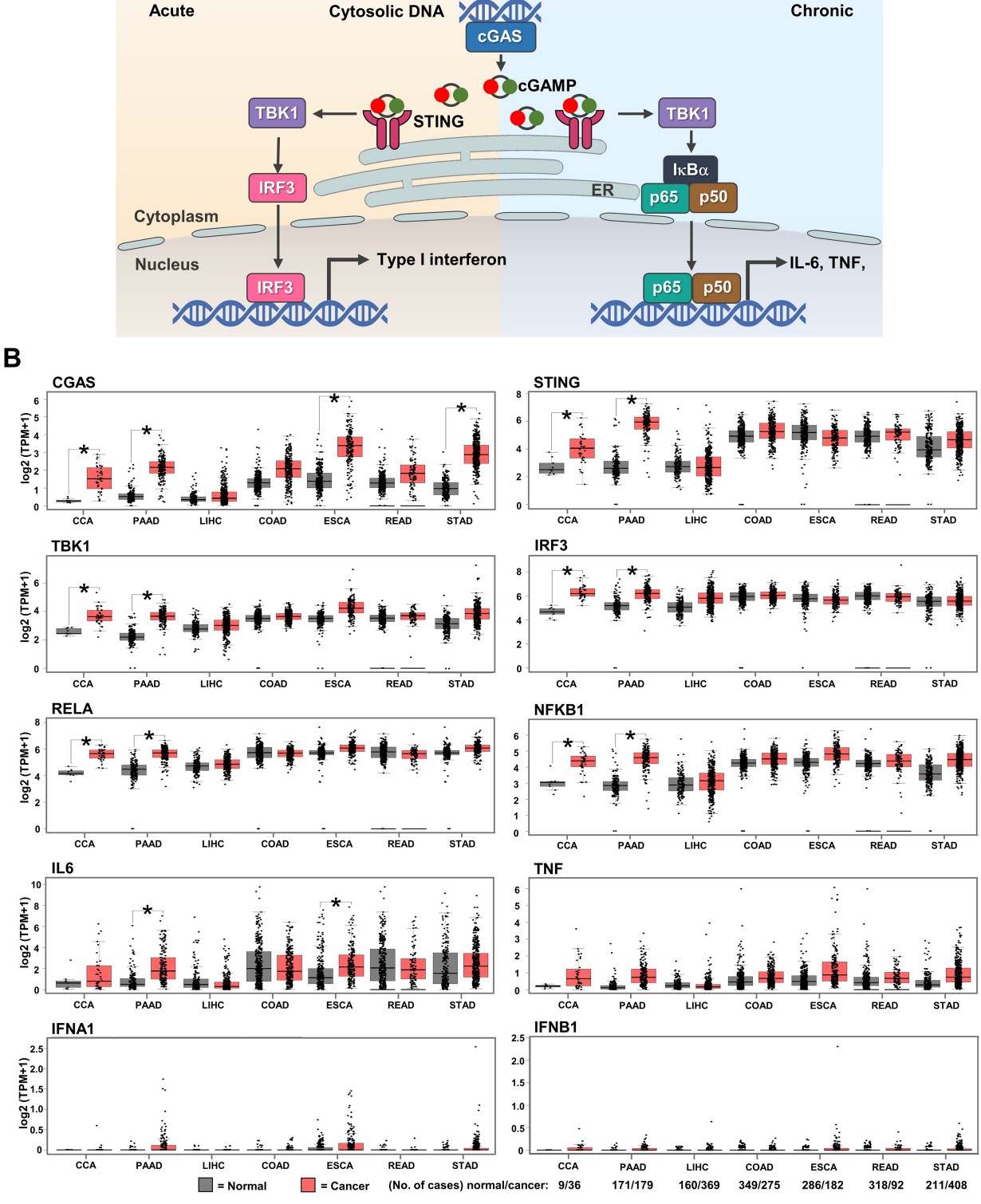

**Figure 2 The mRNA expression levels of cGAS-STING pathway-related molecules in seven gastrointestinal cancers.** (A) The diagram presents the cGAS-STING signaling pathway in cells in acute and chronic inflammatory responses. (B) mRNA levels of *CGAS, STING, TANK-binding kinase 1* (*TBK1*), *interferon regulatory factor 3* (*IRF3*), *NF-κB p65* (*RELA*), *NF-κB p50* (*NFKB1*), *interleukin-6* (*IL6*), *tumor necrosis factor* (*TNF*), *interferon alpha 1* (*IFNA1*), and *IFN beta 1* (*IFNB1*) in cancer tissues (■) compared to normal tissues (■). CCA, pancreatic adenocarcinoma (PAAD), liver

**Figure 2 (continued)**
hepatocellular carcinoma (LIHC), colon adenocarcinoma (COAD), esophageal carcinoma (ESCA), rectum adenocarcinoma (READ), and stomach adenocarcinoma (STAD) were selected as the representative cancers of the gastrointestinal tract. Tissues mRNA level is presented as $\log_2$ (TPM+1). ER = endoplasmic reticulum, IκBα = Inhibitor kappa B-alpha, IL-6 = interleukin-6, TNF = tumor necrosis factor, TPM = transcript count per million. $^*p < 0.05$. Total numbers of cancer and normal samples are presented at the right lower corner. Image credit: GEPIA, http://gepia.cancer-pku.cn/index.html.

**Table 1 Correlations of cGAS, STING and NF-κB p65 expression levels with clinicopathological characteristics of CCA patients.**

| Variables | n (%) | cGAS | | | STING | | | NF-κB p65 | | |
|---|---|---|---|---|---|---|---|---|---|---|
| | | Low | High | $p^{\#}$ | Low | High | $p^{\#}$ | Low | High | $p^{\#}$ |
| **Age (year)** | | | | | | | | | | |
| <56 | 47 (49.5) | 22 | 25 | 0.914 | 24 | 23 | 0.921 | 24 | 23 | 0.759 |
| ≥56 | 48 (50.5) | 23 | 25 | | 25 | 23 | | 23 | 25 | |
| **Gender** | | | | | | | | | | |
| Male | 62 (65.3) | 30 | 32 | 0.785 | 34 | 28 | 0.384 | 30 | 32 | 0.772 |
| Female | 33 (34.7) | 15 | 18 | | 15 | 18 | | 17 | 16 | |
| **Anatomical subtype** | | | | | | | | | | |
| Perihilar CCA | 33 (34.7) | 11 | 22 | 0.046* | 12 | 21 | 0.030* | 19 | 14 | 0.249 |
| Intrahepatic CCA | 62 (65.3) | 34 | 28 | | 37 | 25 | | 28 | 34 | |
| **Histological type** | | | | | | | | | | |
| Papillary | 43 (45.3) | 19 | 24 | 0.572 | 21 | 22 | 0.627 | 21 | 22 | 0.910 |
| Non-papillary | 52 (54.7) | 26 | 26 | | 28 | 24 | | 26 | 26 | |
| **Tumor stage (7th AJCC##)** | | | | | | | | | | |
| I–III | 43 (45.3) | 19 | 24 | 0.572 | 22 | 21 | 0.941 | 22 | 21 | 0.765 |
| IV | 52 (54.7) | 26 | 26 | | 27 | 25 | | 25 | 27 | |
| **Tumor size (n = 78†)** | | | | | | | | | | |
| <7 cm | 37 (47.4) | 20 | 17 | 0.370 | 19 | 18 | 0.821 | 22 | 15 | 0.043* |
| ≥7 cm | 41 (52.6) | 18 | 23 | | 20 | 21 | | 15 | 26 | |

Notes:
\# p-values were determined by Pearson Chi-square test.
\#\# Tumor stage is classified according to the 7th AJCC (*Edge & Compton, 2010*).
\* p < 0.05.
† Incomplete information.

proteins in papillary CCA (Figs. S3A–S3C), were not significantly associated with patient survival. It is also noteworthy that when CCAs were classified anatomically into intrahepatic and perihilar subtypes (Figs. S4–S6 and Table S1–S4), stage IV disease in the perihilar subtype was associated with shorter survival (Fig. S4C). No other clinical parameters showed a significant correlation with patient survival (Figs. S4–S6 and Table S1–S4).

## Non-papillary subtype and high NF-κB p65 are the poor prognostic factors

The multivariate Cox regression analysis indicated that both non-papillary subtype and high NF-κB p65 levels were independent poor prognostic factors of CCA patients (Table 2,
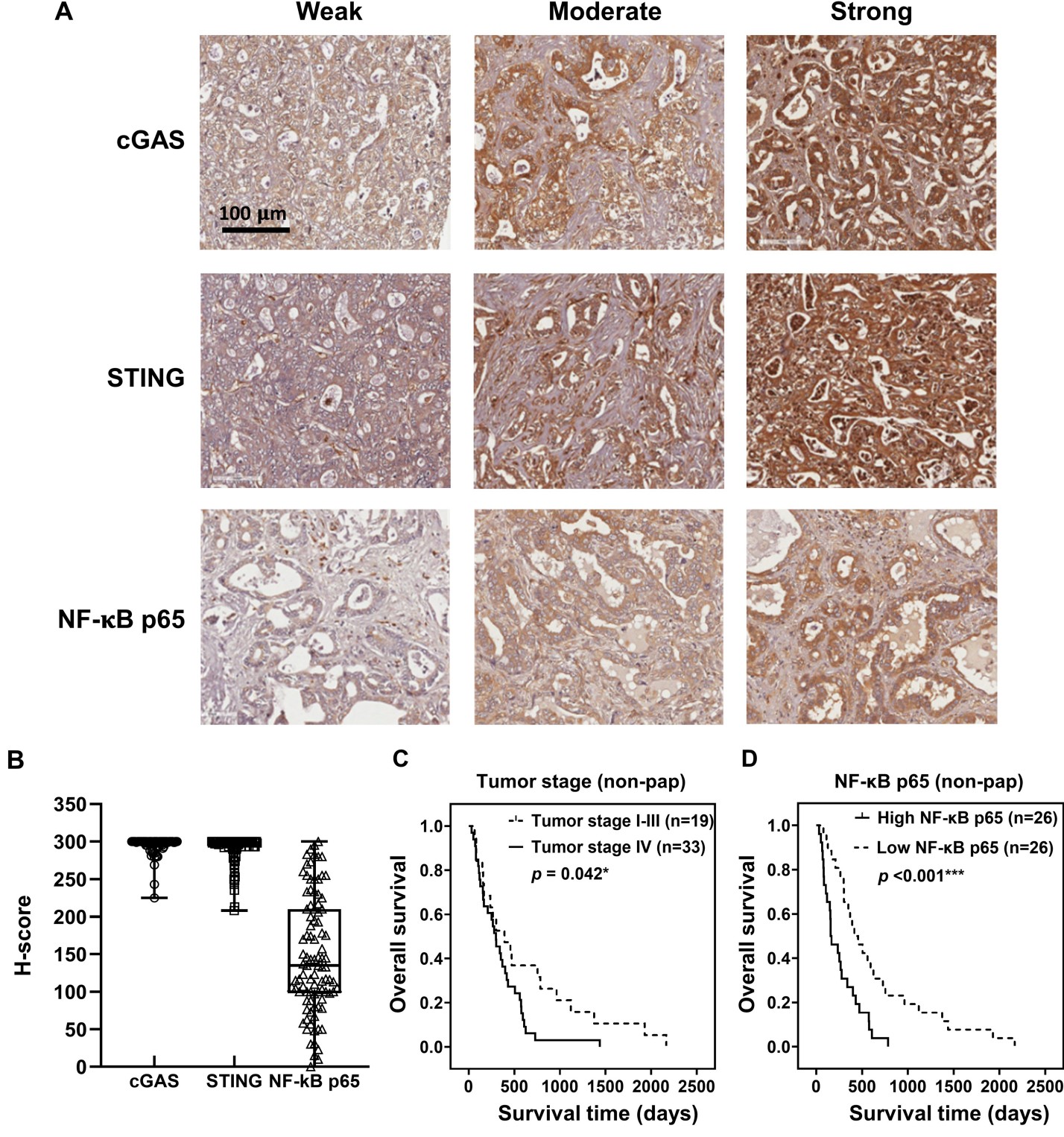

**Figure 3 cGAS, STING, and NF-κB p65 expression in CCA tissues.** (A) The representative immunohistochemistry staining of cGAS, STING, and NF-κB p65 in CCA tissues are categorized into weak, moderate, and strong. Bar = 100 μm. (B) The distributions of cGAS, STING, and NF-κB p65 levels were evaluated by H-score. Kaplan-Meier survival analysis of non-papillary CCA patients (non-pap) with different (C) tumor stages and (D) NF-κB p65 expression. $^*p < 0.05$, $^{***}p < 0.001$.

**Table 2 Multivariate analysis using Cox regression of CCA patients' clinicopathological characteristics.**

| Variables | | $n$ | HR | 95%CI | $p$ |
|---|---|---|---|---|---|
| Age (years) | <56 | 47 | 1 | | |
| | ≥56 | 48 | 1.084 | [0.717–1.639] | 0.704 |
| Gender | Male | 62 | 1 | | |
| | Female | 33 | 1.029 | [0.655–1.618] | 0.900 |
| Anatomical subtype | Perihilar CCA | 33 | 1 | | |
| | Intrahepatic CCA | 62 | 1.017 | [0.641–1.614] | 0.942 |
| Histological type | Papillary | 43 | 1 | | |
| | Non-papillary | 52 | 1.597 | [1.028–2.480] | 0.037* |
| Tumor stage | I–III | 43 | 1 | | |
| | IV | 52 | 1.311 | [0.843–2.038] | 0.229 |
| cGAS expression | <300 | 45 | 1 | | |
| | =300 | 50 | 0.769 | [0.474–1.247] | 0.287 |
| STING expression | <300 | 49 | 1 | | |
| | =300 | 46 | 0.899 | [0.555–1.456] | 0.665 |
| NF-κB p65 expression | <135 | 47 | 1 | | |
| | ≥135 | 48 | 1.679 | [1.074–2.626] | 0.023* |

Notes:
HR, Hazard ratio; CI, Confidence interval.
* $p < 0.05$.

HR = 1.597; 95% CI [1.028–2.480], $p$ = 0.037; HR = 1.679; 95% CI [1.074–2.626], $p$ = 0.023). These finding suggest that both the non-papillary subtype and NF-κB p65 overexpression contribute to poorer prognosis in CCA patients.

### CGAS, STING, and RELA (NF-κB p65) are commonly overexpressed in CCA

To validate our findings, expression levels of *CGAS, STING,* and *RELA* were analyzed using publicly available transcriptomic datasets (*Andersen et al., 2012; Chaisaingmongkol et al., 2017*). Comparison between CCA samples and normal tissues revealed that all three genes were significantly upregulated in CCA (Fig. 4A). When the correlation between two variables was determined by Pearson correlation coefficient, a significant positive correlation was observed between *CGAS* and *STING* expression in both cohorts (r = 0.2682, $p$ = 0.006 and r = 0.2990, $p$ = 0.004). Additionally, *RELA* levels were positively correlated with *STING* expression in the GSE76297 cohort (r = 0.2165, $p$ = 0.039) (Fig. 4B). However, no significant correlation was found between *RELA* and *CGAS* in either dataset.

### Correlations between *CGAS, STING,* and *RELA* expression levels and specific immune cell infiltration (ICI)

To explore the relationship between the expression levels of *CGAS, STING,* and *RELA* were associated with specific ICI, the TIMER2.0 web platform was utilized. The cGAS-STING pathway plays dual roles in promoting both anti-tumor and pro-tumor immunity, which can vary depending on the cancer type and stage (*Khoo & Chen, 2018; Ng et al., 2018;*

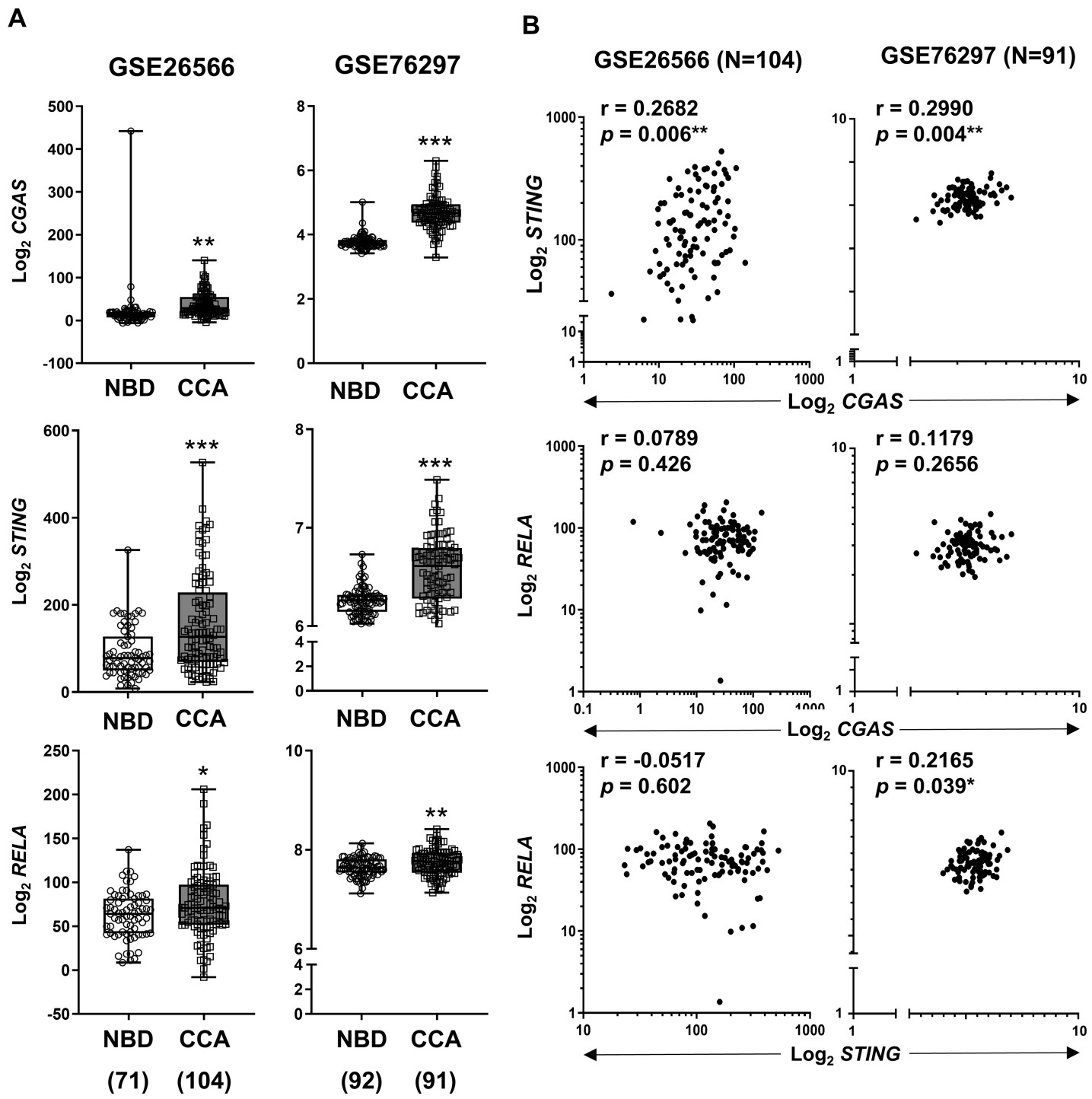

**Figure 4** The mRNA expression levels of CGAS, STING, and NF-κB p65 (RELA) in CCA from two publicly available datasets (GSE26566 and GSE76297). (A) Graphs present mRNA levels in normal bile duct (NBD) and CCA tissues. mRNA levels are presented as log₂ expression. (B) The correlation between two distinct mRNA levels. r = Pearson correlation coefficient. Numbers in brackets present the total number of cases in each group. *$p < 0.05$, **$p < 0.01$, ***$p < 0.001$. Image credit: GEO: https://www.ncbi.nlm.nih.gov/gds/.

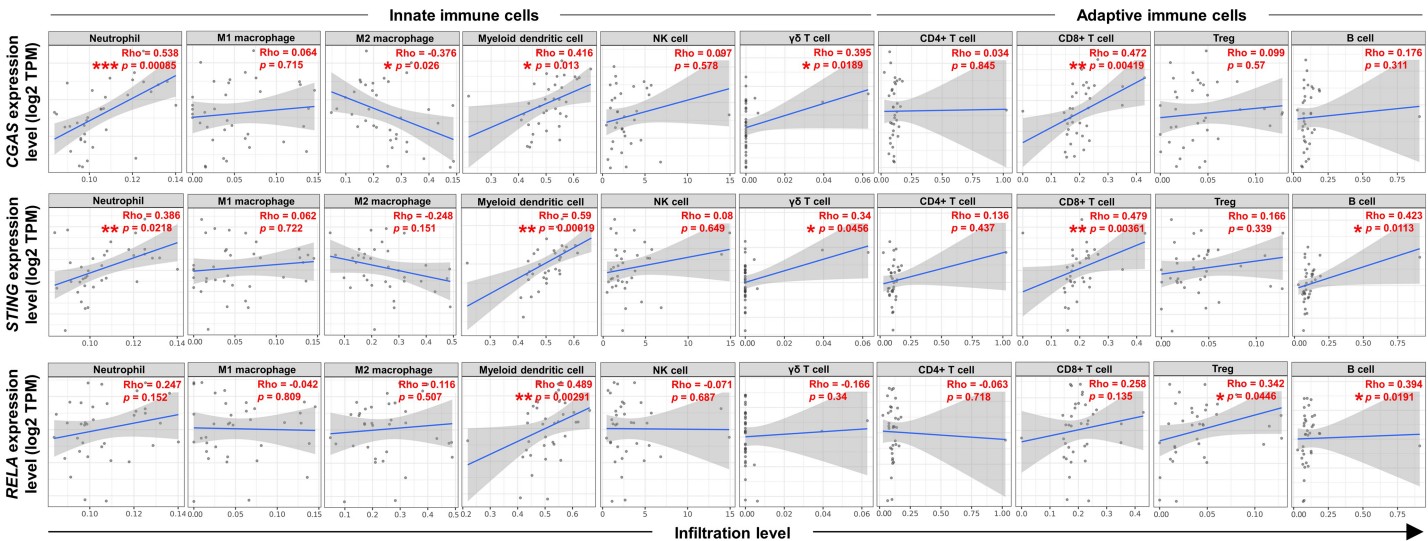

**Figure 5 The correlations between CGAS, STING, and RELA expression levels and ICI.** A scatterplot showed the correlations between *CGAS*, *STING*, and *RELA* expression levels in log2 Transcripts Per Million (TPM) and infiltration levels of immune cells. Data were analyzed using Tumor Immune Estimation Resource (TIMER) 2.0 based on The Cancer Genome Atlas (TCGA) database. The infiltrating immune cells, including neutrophils, M1 and M2 macrophages, myeloid dendritic cells, NK cells, gamma delta (γδ) T cells, CD4+ T cells, CD8+ T cells, regulatory T (Treg) cells, and B cells were demonstrated. The correlation coefficient was shown by the Rho value. A *p*-value indicates statistical significance. $^{*}p < 0.05$, $^{**}p < 0.01$, $^{***}p < 0.001$. Image credit: TIMER2.0, http://timer.cistrome.org/.

*Zheng et al., 2020*). A range of relevant immune cell types engaged in the cGAS-STING immune response were analyzed, including neutrophils (*Nagata et al., 2021*), macrophages (M1 and M2 subtypes) (*Hu et al., 2021*), myeloid dendritic (DC) cells, natural killer (NK) cells, gamma delta (γδ) T cells, CD4+ T and CD8+ T cells, regulatory T (Treg) cells, and B cells (*Khoo & Chen, 2018*; *Liu et al., 2023*; *Luo et al., 2025*; *Ng et al., 2018*; *Zheng et al., 2020*) (Figs. 5). The results showed positive correlations between *CGAS-STING* and the infiltration of neutrophils, γδ T cells, and CD8+ T cells. Conversely, *CGAS* was negatively correlated with M2 macrophages, a cell type typically associated with tumor promotion. *STING* and *RELA* levels were positively correlated with B cell infiltration. Notably, only *RELA* showed a significant positive correlation with immunosuppressive Tregs infiltration. All three genes, *CGAS, STING*, and *RELA*, were positively associated with DC cell infiltration. However, no significant associations were observed between their expression and the infiltration of M1 macrophages, NK cells, or CD4+ T cells. These findings suggest that *CGAS* and *STING* may be linked to anti-tumor immunity, whereas *RELA* might contribute to immunosuppressive mechanisms through its association with Treg infiltration.

# DISCUSSION

CCA is an aggressive cancer originating from the bile ducts, characterized by limited treatment options and a poor prognosis (*Banales et al., 2020*; *Vogel et al., 2023*). The identification of novel therapeutic targets and prognostic markers is critical for improving patient outcomes. In this context, the cGAS-STING pathway is under active investigation

for its role in cancer-related immune responses. In the present study, we observed the presence of micronuclei (MNs), known inducers of cGAS, in CCA cell lines, together with a positive correlation between MNs and STING expression. The clinical relevance of selected molecules in the cGAS-STING pathway was further explored using publicly available transcriptomic data from the TCGA database. Most associated mRNAs were significantly upregulated in both CCA and PAAD samples, suggesting potential roles in these cancer progressions. Immunohistochemical analysis of Thai CCA tissues revealed frequent high expression of cGAS and STING and identified NF-κB p65 as a potential prognostic marker. Transcriptomic data from the GEO database further supported the elevated expression of *CGAS*, *STING*, and *RELA* in CCA tissues compared to normal bile ducts. TIMER2.0 analysis proposed the potential involvement of the cGAS-STING-NF-κB axis in the immune modulation in CCA. ICI analysis revealed that *CGAS* and *STING* expression levels positively correlated with infiltrating neutrophils, cytotoxic CD8$^+$ T cells, and γδ T cells, while DC cells were elevated in CCA with increased expression of *CGAS*, *STING*, and *RELA*. Collectively, these findings suggest that the cGAS-STING pathway may serve as a promising therapeutic target in CCA and that NF-κB p65 may function as a relevant prognostic biomarker.

Chromosomal instability (CIN) is a hallmark of cancer, predisposing cells to errors during mitosis that often result in the formation of MNs (*Crasta et al., 2012*; *Tanaka & Hirota, 2016*). MNs have been linked to the innate immune response triggering *via* the cGAS-STING signaling pathway (*Dou et al., 2017*; *Gluck et al., 2017*; *Harding et al., 2017*; *Mackenzie et al., 2017*). The current study demonstrated the MNs in CCA cell lines and their generation during the chromosomal segregation. The numbers of CCA MNs correlated with STING levels. The paclitaxel-induced MN formation led to the cGAS-STING pathway activation was previously demonstrated in triple-negative breast cancer (*Hu et al., 2021*). Additionally, cGAS-independent activation of STING has been reported, involving p53 and E3 ubiquitin ligase TRAF6 (*Dunphy et al., 2018*). The wild-type p53 in KKK-D068 (*Vaeteewoottacharn et al., 2019*) and an inactive mutant form of p53 in KKU-213A (or formerly known as M213) (*Phimsen et al., 2012*) suggest that the interplay between cGAS, STING, and p53 in MN-induced activation may differ among CCA cell lines and warrants further investigation.

The cGAS-STING pathway is an evolutionarily conserved self-defense mechanism, that detects cytosolic DNA and mediates immune activation (*Mackenzie et al., 2017*; *Zheng et al., 2020*). The cGAS-STING pathway was reported to have dual effects in cancers (*Bakhoum & Cantley, 2018*; *Khoo & Chen, 2018*; *Ng et al., 2018*). It has strong anti-tumor immunity and enhances the effectiveness of various anti-cancer treatments; however, in specific types of cancers, the cGAS-STING pathway contributes to cancer growth and metastasis by altering the cancer microenvironment. For instance, STING activation induces immunosuppressive cytokines in HPV-related tongue squamous cell carcinoma (*Liang et al., 2015*), but promotes anti-tumor immunity in cervical cancer (*Shi et al., 2021*). Our findings of elevated cGAS and STING mRNA levels in CCA and PAAD are consistent with previous reports (*Kabashima et al., 2022*; *Luo et al., 2024*; *Sun et al., 2024*). Although high expression of cGAS and STING was frequently observed in perihilar CCA, their

clinical implications were not demonstrated in our cohort. Nevertheless, the elevated expression of these molecules may offer an opportunity for CCA treatment using cGAS or STING agonists (*Kabashima et al., 2022*; *Luo et al., 2024*).

It is well established that the canonical activation of cGAS-STING leads to TBK1 stimulation and type I IFN production, or alternatively, to IKK-mediated activation of NF-κB p65-p50 and proinflammatory cytokine production (*Khoo & Chen, 2018*; *Ng et al., 2018*). While NF-κB plays dual roles in cancer, acting as both a cancer suppressor and promoter (*Lalle, Twardowski & Grinberg-Bleyer, 2021*), accumulating evidence indicates that its activation in CCA promotes cancer progression and may serve as a potential therapeutic target (*Seubwai et al., 2014*). Although previous studies have noted elevated levels of NF-κB in CCA (*Seubwai et al., 2014*), our study is the first to demonstrate its prognostic relevance. NF-κB p65 levels positively correlated with tumor size, and higher expression was significantly associated with shorter patient survival. The difference in the diagnostic power of NF-κB between studies may be attributed to sample size differences; analyzed 48 samples, whereas the current study included 95 cases (*Seubwai et al., 2014*). Previous studies have shown the therapeutic potential of targeting NF-κB in CCA (*Seubwai et al., 2010*; *Seubwai et al., 2014*). Our finding suggested that NF-κB p65 is not only a candidate therapeutic target but also a potential prognostic biomarker. A larger cohort may be required to establish a clinically relevant cut-off value for NF-κB p65 expression.

The cGAS-STING pathway also plays a central role in immune modulation (*Ng et al., 2018*). The activation of cGAS-STING promotes neutrophil accumulation and enhances CD8[+] T cell-mediated anti-tumor immunity in breast cancer and melanoma (*Nagata et al., 2021*). STING activation also mediates γδ T cell responses in melanoma (*Luo et al., 2025*). In the present study, the ICI analysis revealed that *CGAS* and *STING* expression correlated positively with neutrophils, CD8[+] T cells, and γδ T cells in CCA. Although a previous study demonstrated that cGAS can promote M1 polarization in breast cancer (*Hu et al., 2021*), our data did not show significant associations between *CGAS, STING, or RELA* and M1 macrophage infiltration. However, a negative correlation was observed between *CGAS* and M2 macrophages, supporting a potential anti-tumor role. Furthermore, consistent with the role of the cGAS-STING pathway in bridging innate and adaptive immunity *via* DCs (*Ng et al., 2018*; *Zheng et al., 2020*), we observed positive associations with DC infiltration. Interestingly, only RELA expression correlated with Tregs, which have been implicated in CCA progression and metastasis (*Zhang et al., 2023*).

While our findings propose the clinical relevance of the cGAS-STING-NF-κB axis as both a therapeutic and prognostic target in CCA, these hypotheses require validation in additional experimental models.

## CONCLUSIONS

High expression of cGAS and STING in CCA tissues suggests their potential as actionable targets for CCA treatment. Additionally, elevated NF-κB p65 expression demonstrated clinical relevance as a prognostic marker. The association of these molecules with specific immune cell populations, including neutrophils, T cells, and antigen-presenting cells,

highlights their potential roles in modulating the tumor immune microenvironment. Further studies are mandatory to clarify the immunological functions of the cGAS-STING pathway in CCA and to explore the therapeutic potential of targeting cGAS-, STING-, or NF-κB-related molecules.

## ACKNOWLEDGEMENTS

We would like to thank the Cholangiocarcinoma Research Institute (CARI), Thailand for providing CCA tissues.

### Funding

This study was supported by the Fundamental Fund of Khon Kaen University, Thailand, and the Faculty of Medicine, Khon Kaen University, Thailand (Grant Number IN65205). Han Ni Zin Myint was supported by the KKU Active Recruitment Project 2022 and Teaching Assistant (TA) Scholarship for the Academic Year 2024. The funders had no role in study design, data collection and analysis, decision to publish, or preparation of the manuscript.

### Grant Disclosures

The following grant information was disclosed by the authors:
Faculty of Medicine, Khon Kaen University, Thailand: IN65205.
KKU Active Recruitment Project 2022 and Teaching Assistant (TA) Scholarship for the Academic Year 2024.

### Competing Interests

The authors declare that they have no competing interests.

### Author Contributions

- Han Ni Zin Myint conceived and designed the experiments, performed the experiments, analyzed the data, prepared figures and/or tables, authored or reviewed drafts of the article, and approved the final draft.
- Paksiree Saranaruk conceived and designed the experiments, performed the experiments, analyzed the data, prepared figures and/or tables, authored or reviewed drafts of the article, and approved the final draft.
- Natnicha Paungpan conceived and designed the experiments, performed the experiments, analyzed the data, prepared figures and/or tables, authored or reviewed drafts of the article, and approved the final draft.
- Sakda Waraasawapati conceived and designed the experiments, analyzed the data, authored or reviewed drafts of the article, and approved the final draft.
- Chawalit Pairojkul conceived and designed the experiments, analyzed the data, authored or reviewed drafts of the article, and approved the final draft.
- Saowaluk Saisomboon conceived and designed the experiments, performed the experiments, authored or reviewed drafts of the article, and approved the final draft.

- Nutsara Settha conceived and designed the experiments, performed the experiments, authored or reviewed drafts of the article, and approved the final draft.
- Jatuporn Pattanachot conceived and designed the experiments, performed the experiments, authored or reviewed drafts of the article, and approved the final draft.
- Seiji Okada conceived and designed the experiments, authored or reviewed drafts of the article, providing some resources, and approved the final draft.
- Wunchana Seubwai conceived and designed the experiments, authored or reviewed drafts of the article, providing some resources, and approved the final draft.
- Amnat Kitkhuandee conceived and designed the experiments, authored or reviewed drafts of the article, providing some resources, and approved the final draft.
- Kanlayanee Sawanyawisuth conceived and designed the experiments, authored or reviewed drafts of the article, and approved the final draft.
- Kulthida Vaeteewoottacharn conceived and designed the experiments, prepared figures and/or tables, authored or reviewed drafts of the article, and approved the final draft.

## Human Ethics

The following information was supplied relating to ethical approvals (*i.e.*, approving body and any reference numbers):

The study procedure was approved by the Ethics Committee for Human Research of Khon Kaen University based on the Declaration of Helsinki (HE641574 and HE651124).

## Data Availability

The raw measurements are available in the Supplemental Figures and Tables.

## Supplemental Information

Supplemental information for this article can be found online at http://dx.doi.org/10.7717/peerj.19800#supplemental-information.

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
