# Peer review of "High cyclic GMP-AMP synthase and stimulator of interferon genes in cholangiocarcinoma suggest their potential as targets for treatment"

_PeerJ, doi:10.7717/peerj.19800_

## Round 0.1 · original submission · Major Revisions

Please address comments given by all 3 reviewers and provide a response to reviewers in a point-wise manner.

Reviewer 1 ·

Basic reporting

The manuscript by Myint et al. explores the role of the cGAS-STING pathway in cholangiocarcinoma and provides important insights into this area. The authors utilized their cholangiocarcinoma cohort (n = 95), conducted in vitro experiments, and complemented their findings with publicly available data. Overall, the study is scientifically accurate and presents novel findings. However, the following comments need to be addressed to improve the narrative, support the findings, and provide clarity.

1. Including a brief overview of the NF-kB and cGAS-STING pathways in the introductory section could be valuable for readers, especially emphasizing the link between the two pathways.

2. Lines 83-85: As stated in the manuscript, chronic NF-kB activation and the resulting inflammation have been linked with carcinogenesis. However, it may also be valuable to acknowledge the literature surrounding NF-kB-mediated immune activation, which may be anti-tumorigenic. Reference highlighting the “dual role” of NF-kB: Twardowski, J., & Grinberg-Bleyer, Y. (2021). NF-κB in cancer immunity: friend or foe?. Cells, 10(2), 355.

3. Lines 89-92: When discussing standard-of care, it would be valuable to add information and a corresponding reference about gemcitabine/cisplatin with PD-L1 inhibitors as first-line therapy for advanced disease. (Keynote-966 and Topaz-1 studies). Since the text states that surgery is the “gold standard treatment,” it would also be valuable to mention the percent of patients diagnosed with metastatic vs. surgically treatable disease.

Experimental design

1. Lines 198-208: When analyzing micronuclei in CCA cell lines, it would be valuable to have a cell line derived from normal tissue or use existing cell line for comparison. In the corresponding figure 1B, it is unclear what the two groups the statistically significant comparison is between. Further explanation of the y-axis would also be beneficial (denominator of the percentage is unclear).

Validity of the findings

1. Line 238: If possible, it could be valuable to include separate Western blots for nuclear and cytoplasmic lysates to account for the importance of NF-kB cytoplasm-to-nucleus translocation.

Additional comments

1. Lines 235 – 236: The text stats that “the overall survival times calculated from surgery to death ranged from 30 to 2,593 days, with a median survival time of 377.” Does this mean that all patients included in the study received surgery? This is surprising since line 233 noted that 54.7% of patients had stage IV metastatic disease.

2. Lines 227 -236: Is there any information available about whether patients had cholangiocarcinoma tumors associated with primary sclerosing cholangitis or liver fluke infection? These well-known risk factors could be relevant to NF-kB and cGAS-STING signaling explored in the manuscript.

3. Is immune data available for any of the samples or publicly available datasets? It could be interesting/relevant to examine relationships between NF-kB, cGAS/STING, and immune cell infiltration.

Annotated reviews are not available for download in order to protect the identity of reviewers who chose to remain anonymous.

Reviewer 2 ·

Basic reporting

The authors have made very good efforts in writing this research article High cGAS and STING in cholangiocarcinoma imply their potential as targets for treatment, but there is room for improvement.
• Some sentences are very long and complex. Breaking these up could improve readability.
• What is the rationale of selecting High cGAS and STING in cholangiocarcinoma
• The introduction needs to be started with cancer overview
• The conclusion part is very concise, lacks depth and is disorganized, it needs further elaboration
• There are plenty of grammatical mistakes, so it needs proper language editing.
• I don't feel this is a timely review due to its referencing. Ensuring a uniform and accurate citation style is indeed crucial for maintaining the quality of manuscript.

Experimental design

No Comments

Validity of the findings

No comments

Additional comments

No Comments

Reviewer 3 ·

Basic reporting

Strengths:

The manuscript is written in professional and clear English, suitable for an international scientific audience.
The introduction effectively highlights the importance of chromosomal instability (CIN) and the cGAS-STING pathway in cancer biology, with a focus on the poorly understood role in cholangiocarcinoma (CCA).
Figures and tables are relevant, well-annotated, and visually clear, aiding comprehension of the results.
Critiques:

Introduction:

The introduction frames the role of cGAS-STING in cancer but lacks a thorough discussion of opposing literature that questions its therapeutic potential. For example, in some cancers, cGAS-STING signaling supports tumor progression through inflammatory cytokines. This nuance needs better articulation.
There is insufficient discussion of how this study specifically advances the field compared to prior findings in CCA or other gastrointestinal cancers.
Clarity of Results:

Phrases like “virtually highly expressed” (line 58) and “importance of MNs” (line 63) are vague and imprecise. These should be revised for scientific clarity.
The Kaplan-Meier survival analysis shows differential expression correlations but does not explicitly explain why cGAS and STING, despite being overexpressed, fail to correlate with survival. This contradiction needs further elaboration.
References and Context:

Although the manuscript is well-referenced, the inclusion of newer studies on cGAS-STING in immune-oncology could better situate the findings within the current landscape of cancer therapeutics.
The rationale for selecting NF-κB p65 as the key prognostic marker is not thoroughly justified in the introduction or discussion.

Experimental design

Strengths:

The combination of in vitro and ex vivo data enhances the manuscript’s scientific robustness.
Ethical compliance is well-documented, including approval for human tissue use and cell line provenance.
Critiques:

Methodological Transparency:

The study does not sufficiently detail how expression levels were categorized into “low” and “high” groups for statistical analyses. Were median H-scores or predefined thresholds used? These details are crucial for reproducibility.
The selection criteria for clinical samples (95 CCA tissues) should be clarified. Are the samples representative of the CCA patient population in terms of stage, subtype, or anatomical location?
Sample Size and Power:

The sample size for certain analyses (e.g., subtype-specific survival) appears underpowered. For instance, splitting into papillary and non-papillary subtypes might introduce statistical artifacts. The authors should address this limitation directly or provide a power analysis.
Biological Mechanisms:

The study observes high cGAS and STING expression but does not functionally validate their roles in CCA progression. This weakens the assertion that these are actionable therapeutic targets. Incorporating experiments using cGAS/STING agonists or antagonists in vitro or in vivo would substantiate these claims.
Statistical Analyses:

The statistical analyses, while adequate, could be expanded to include multivariate analyses for cGAS and STING alongside NF-κB p65 to uncover potential interactive effects.
Cox regression models are well-utilized but should explicitly address whether confounding factors (e.g., tumor stage, subtype, treatment history) were adequately controlled.

Validity of the findings

Strengths:
The study identifies NF-κB p65 as a prognostic marker, a potentially novel finding in CCA.
The use of publicly available datasets (GEO and TCGA) lends additional validity to the observations of overexpressed cGAS-STING-related molecules in gastrointestinal cancers.
Critiques:

Weakness in Causal Assertions:

The manuscript highlights a correlation between micronuclei (MNs) and STING levels but does not explore causality. For instance, are MNs directly activating cGAS-STING signaling? Could the authors employ STING knockdown or pharmacological inhibition to establish this link?
While NF-κB p65 is linked to poorer prognosis, the mechanistic basis for this association is underexplored. What downstream pathways or cytokines are driving this effect?
Confounding Variables:

The study’s reliance on H-scores and categorical groupings for protein expression introduces potential biases. For example, intrahepatic CCA (iCCA) may have inherently different biology compared to perihilar CCA, which could confound the findings. Stratified analyses are recommended.
Therapeutic Implications:

The conclusion that cGAS-STING are therapeutic targets is speculative without functional data. The authors should temper this assertion or provide preliminary validation, such as testing cGAS/STING pathway inhibitors or agonists in CCA models.
Integration of Data Sources:

Discrepancies between the expression patterns observed in the authors’ IHC data and publicly available datasets should be better reconciled. Are methodological differences responsible, or do these reflect true biological variability?

Additional comments

The manuscript presents intriguing findings on the cGAS-STING pathway and its association with prognostic markers in CCA. However, the conclusions, particularly regarding therapeutic targeting, are premature without further mechanistic exploration. The work would significantly benefit from additional experiments to:

Validate cGAS-STING activation using agonists/antagonists.
Investigate downstream signaling and cytokine profiles in relation to NF-κB p65 expression.
Expand on the clinical relevance of the cGAS-STING pathway beyond correlative evidence.
Additionally, while the study provides valuable insights, its generalizability is limited by the focus on Thai CCA tissues. Including data from diverse cohorts or discussing the broader applicability of findings would enhance its impact.

---

## Round 0.2 · Minor Revisions

Please address comments given by Reviewer 1 and provide a response to reviewers in a point-wise manner.

Reviewer 1 ·

Basic reporting

Authors have appropriately addressed most of the comments.

Experimental design

Authors have appropriately addressed the comments

Validity of the findings

Authors have appropriately addressed the comments

Additional comments

Authors have not fully addressed this comment:
Lines 83-85: As stated in the manuscript, chronic NF-kB activation and the resulting inflammation have been linked with carcinogenesis. However, it may also be valuable to acknowledge the literature surrounding NF-kB-mediated immune activation, which may be anti-tumorigenic. Reference highlighting the “dual role” of NF-kB: Twardowski, J., & Grinberg-Bleyer, Y. (2021). NF-κB in cancer immunity: friend or foe?. Cells, 10(2), 355.”


It is highly recommended to include references to ensure that the findings do not appear to ignore previously published data in this context.

---

## Round 0.3 · accepted · Accept

Authors have addressed all of the reviewers' comments and the manuscript is ready for publication.